# Medium-Term Comparative Effects of Prescribed Burning and Mechanical Shredding on Soil Characteristics in Heathland and Shrubland Habitats: Insights from a Protected Natural Area

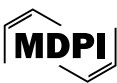

Rosa M. Cadenas [1], Fernando Castedo-Dorado [2] and Luz Valbuena [1,*]

[1] Area of Ecology, Department of Biodiversity and Environmental Management, University of León, Campus of Ponferrada, 24071 León, Spain; rosi.cadenas@gmail.com or cadferro@jcyl.es

[2] Department of Engineering and Agricultural Sciences, University of León, Campus of Ponferrada, 24401 Ponferrada, Spain; fcasd@unileon.es

* Correspondence: luz.valbuena@unileon.es

**Abstract:** Parts of the Cantabrian Mountains (N Spain) have been colonized by woody species in the past six or seven decades as a result of a decline in livestock activity and changes in the fire regime. Various management strategies have been used to prevent the expansion of shrubs and recover grassland ecosystems for grazing activities. However, it is not clear how different vegetation treatments affect soils, which are crucial in supporting life and providing nutrients in these ecosystems. The aim of the present study was to compare the dynamics of the physicochemical and biological soil properties after two vegetation treatments: prescribed burning and shredding. Samples were obtained from plots representing alkaline and acidic soils dominated by gorse shrub (*Genista hispanica* subsp. *occidentalis*) and heath (*Calluna vulgaris*) plant communities, respectively. The soil samples were collected immediately before and after the treatments and one and two years later. The level of available P varied depending on the soil pH, and it only increased after the treatments in the acidic soils in the heathland community. The total N and available P concentrations were higher after the prescribed burning, and the enzymatic activity tended to be higher after the shredding treatment. Despite the significant effects on some soil variables, prescribed burning and shredding did not have important short- and medium-term effects on the chemical and soil enzymatic properties. These treatments can therefore be considered sustainable vegetation management tools, at least in the medium term.

**Keywords:** soil properties; prescribed fire; mechanical shredding; gorse shrubland; heathland; soil nutrients; enzymatic activities

## 1. Introduction

Mountain landscapes and the associated plant cover have been changing over a period of millions of years as a result of the effects of both biotic and abiotic factors. Before the presence of humans, lightning-induced fire was the main agent that affected vegetation and generated dynamic renewal, both in time and space [1]. Since Neolithic times, rural communities in the mountains of Southern Europe have traditionally used fire, shrub clearing and grazing as tools to manage mountain landscapes [2–4]. For example, in Spain, in the Middle Ages, deforestation was carried out to extend summer grasslands for livestock. The practice of moving livestock over long distances in seasonal cycles (transhumance) exacerbated the forest decline.

There is evidence that the abovementioned factors have been key to maintaining open landscapes since Neolithic times in the Cantabrian Mountains (N Spain) [5,6], specifically in the area encompassed by the Babia y Luna Natural Park (an area representative of the Cantabrian Mountains) [7].

Throughout the second half of the 20th century, grasslands underwent a process of decline, largely favored by land use changes and natural succession [8–10] and promoted by attempts to exclude fire and by the abandonment of traditional livestock grazing practices. Additionally, extensive traditional livestock grazing, with low to moderate stocking densities and the short seasonal use of grasslands, benefits the encroachment of both shrubs and trees. Although shrub encroachment can have some positive effects, such as reducing soil erosion and above-ground carbon sequestration, it can also reduce biodiversity and ecopastoral value and can lead to the accumulation of woody fuel, with the associated high fire risk [11].

Several methods of managing heathlands and shrublands have been tested in the Iberian Peninsula. These include conventional methods such as prescribed burning and shrub clearing [12–16], as well as others such as mechanical shredding [16]. According to the Natural Resources Management Plan for the Babia y Luna Natural Park [17], the use of prescribed fire to generate grazing resources is allowed in areas used by transhumant sheep for grazing, while the use of shrub shredding is usually prioritized. Nevertheless, to date, little is known about the effects of these treatments on the ecosystem (and specifically on the soils) and the factors that justify the use of one treatment over the other.

Soil is an invaluable non-renewable resource [18] as it serves as an essential support for natural systems [19]. Soil degradation processes, i.e., the deterioration of the physical, chemical and/or biological properties of soils, are therefore likely to have irreversible consequences. In forest ecosystems, fire and the removal of vegetation cover have been identified as the main causes of soil degradation [20]. Numerous studies have addressed different aspects of the effects of the above-mentioned treatments on soil properties [21–27]. However, the effects of different management practices (vegetation treatments) on the characteristics of different types of soil are less well known (e.g., in the Babia y Luna Natural Park, where both alkaline and acidic soils coexist). Moreover, very few studies [23] have compared the medium-term effects of different vegetation treatments on soil characteristics.

According to the most recent map of crops and natural areas in Castilla y León [28], 27% of the surface area of the Babia y Luna Natural Park corresponds to grassland, 69% to heathland and shrubland and only 1% to woodland. Moreover, a trend towards increasing cover by heathland and shrubland communities has been observed in the past seven decades, mainly caused by the decline in sheep herding along transhumance routes, placing the characteristic large-scale mosaic of open shrublands and grasslands in the Natural Park at risk [29]. This scenario emphasizes both the importance of implementing shrub management practices to halt shrub encroachment, at least in some areas [30], and the need to determine the effects that these practices have on the soil as a key element of the ecosystem.

The main aims of this study were as follows: (i) to compare the temporal changes in selected soil properties after applying prescribed burning and mechanical shredding treatments to two shrub ecosystems with soil of a different pH and (ii) to analyze the effects of these treatments in the medium term, relative to the pre-treatment situation. Our research hypothesis is that prescribed fire and shrub shredding will have similar effects on the physical, chemical and biological properties of both alkaline and acidic soils, and therefore they will be suitable management tools to improve grassland resources while conserving habitats.

## 2. Materials and Methods

### 2.1. Study Area

The study area is located on the southern slope of the Cantabrian Mountain range (NW Spain), in the Babia y Luna Natural Park (Figure 1). The UTM coordinates of the midpoint of the study area are 240,236 m, 4,769,645 m, UTM zone 29N (EPSG: 25829) (Figure 1).

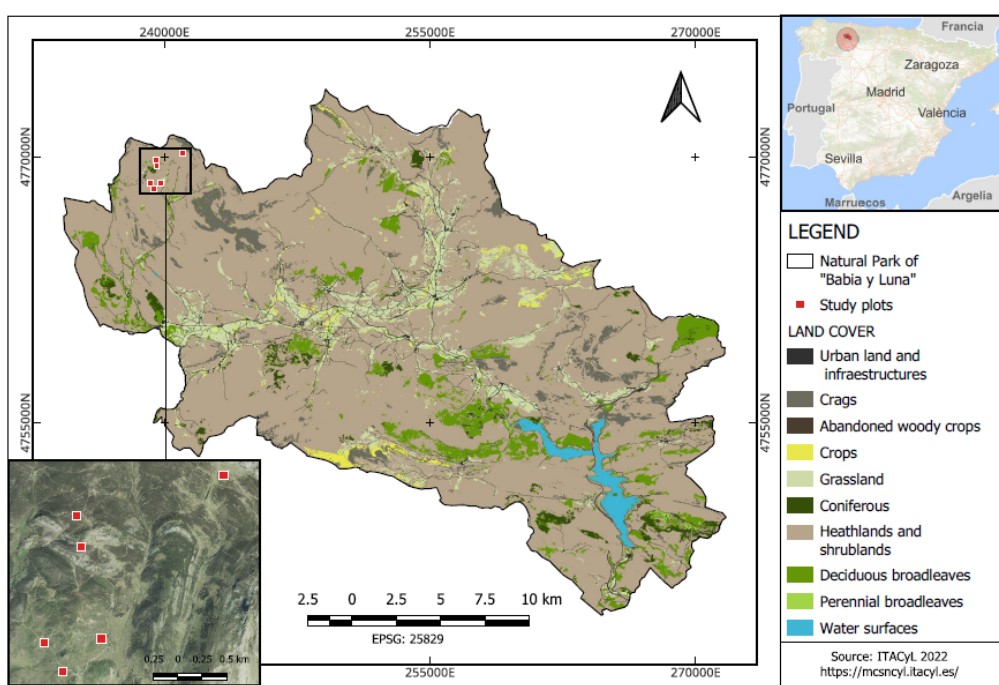

**Figure 1.** Map showing the location of the study area (Babia y Luna Natural Park) in Spain and the spatial distribution of the land use types within the park.

The area is predominantly mountainous, with elevation of more than 1700 m above sea level; the relief is abrupt, except in the valley bottoms. The Babia y Luna Natural Park was declared a Biosphere Reserve by UNESCO in 2004.

According to the Köppen–Geiger climatic classification, the climate in the study area is continental, with a cool or cold summer, no dry season and very cold and snowy winters [31]. The average temperature of the coldest month is below −3 °C (or 0 °C) and that of the warmest month is above 10 °C. The rainfall regime is humid, with adequate rainfall throughout the year, exceeding 1100 mm per year.

Two predominant types of lithology can be distinguished: siliceous zones that produce acidic soils and limestone zones characterized by white or greyish rocks that produce alkaline soils. The differences in soil acidity determine the type of shrub cover. Thus, heathland occurs on acidic soils, with heather (*Calluna vulgaris* (L.) Hull.) predominating, although other species, such as *Daboecia cantabrica* ((Huds.) K. Koch.), *Erica australis* L., *Erica tetralix* L. and *Erica vagans* L., are also common. Gorbizo (*Juniperus sabina* subsp. *alpina* (Suter) Čelak.), mountain broom (*Cytisus oromediterrneus* Rivas Mart. et al.) and blueberry (*Vaccinium myrtillus* L.) may also appear. The vast majority of alkaline soils are occupied by gorse (*Genista hispanica* subsp. *occidentalis* Rouy.).

### 2.2. Vegetation Treatments

The burning and shredding treatments were carried out in October 2016. The treatments were carried out in autumn because, in addition to being able to meet the established prescription window, this is when shepherds traditionally carry out pastoral burning.

The shredding treatment was conducted by a crew belonging to the firefighting prevention and suppression service of the regional government (Junta de Castilla y León). The treatment was conducted with a motorized brush cutter with a shredding blade, with cutting movements from top to bottom. The burning treatment was carried out by the same crew, with support provided by a pumper unit. Prior to the treatments, an area of at least 1 m in width was cleared around the burn areas with a motorized brush cutter. The prescribed burns were carried out on two days (3rd and 4th October, 2016), within a previously established prescription window. The weather conditions on the days that the burns were conducted are summarized in Table 1.

**Table 1.** Mean values and standard deviations (in brackets) of the main meteorological variables during the burnings at the treatment sites with both scrub and heathland.

| Treatment Site | Wind Speed (km/h) | Wind Direction | Temperature (°C) | Relative Humidity (%) |
|---|---|---|---|---|
| **Shrubland** | | | | |
| G1 | 12.72 (5.41) | S | 19.78 (0.80) | 37.41 (1.33) |
| G2 | 14.82 (4.57) | SE | 17.74 (0.62) | 40.12 (1.13) |
| G3 | 18.06(4.12) | SE | 18.53 (0.53) | 40.82 (1.48) |
| **Heathland** | | | | |
| C1 | 17.93 (1.24) | SW | 12.77 (0.90) | 78.57 (3.43) |
| C2 | 6.93 (0.59) | S | 18.43 (0.60) | 51.30 (4.40) |
| C3 | 16.70 (1.65) | W | 10.60 (0.89) | 89.63 (4.76) |

*2.3. Experimental Plots*

The experimental plots were established on hillsides within sites where traditional livestock management involved the short-term grazing of sheep. The experiment was deliberately carried out in areas where the sheep would feed on the tender shoots of re-growing shrubs.

The experimental plots were established in the most representative plant communities of each type of soil to enable a comparison of the effects of burning and shredding treatments on soil properties. Thus, in alkaline soils, the plots were established in shrublands dominated by *Genista hispanica* subsp. *occidentalis*, whereas, on acidic soils, they were established in heathlands dominated by *Calluna vulgaris*. Thirty-six experimental plots were established in six treatment sites: three on scrub (G1, G2, G3) and three in heathland (C1, C2, C3). Six square plots with a size of 10 m were delineated in each of the six treatment sites: three corresponding to the shredding treatment and three to the prescribed burning treatment.

In each of the six plots, five sampling points were established to assess the ground cover. Four points were located two meters from the ends of the plot diagonals, with the remaining point in the center of the plot. At each sampling point, a one-square-meter vegetation quadrat was used to assess the cover of bare soil, rock, scrub and herbaceous vegetation. It is noteworthy that herbaceous plants usually grow under shrubs; hence, the total cover can exceed 100% due to plant overlapping.

The plots in the gorse shrublands included a high percentage of grass cover, whereas the heathlands were characterized by a low percentage of grass and a higher percentage of bare ground (Table 2). According to the Scott and Burgan [32] classification of fuel models, the predominant classes are GS1 (heathland) and GS3 (gorse shrubland).

**Table 2.** Mean values and standard deviations (in brackets) of the main characteristics of the land cover in the experimental plots.

| Land Cover Variable | Shrub-Dominated Plots | Heath-Dominated Plots |
|---|---|---|
| Shrub height (cm) | 25.96 (7.50) | 27.16 (5.68) |
| Shrub cover (%) | 82.66 (20.49) | 85.23 (13.84) |
| Herbaceous cover (%) | 19.17 (18.37) | 3.30 (4.42) |
| Bare ground (%) | 1.76 (3.76) | 9.56 (9.42) |
| Rocks (%) | 0.44 (1.40) | 0.03 (0.31) |

*2.4. Soil Sampling and Analysis*

Soil samples were collected immediately before (pre_0) and immediately after (post_0) the treatments and one year (1) and two years (2) later. Soil samples were extracted with an extraction cylinder (diameter, 7 cm and depth, 3 cm), at five different points in each sampling plot (i.e., 36 × 5 = 180 samples): four samples were taken at a distance of 1 m from the ends of each diagonal of the 10 m sized plots, and the other sample was taken

from the centers of the plots. In the burned plots, the layer of ash was removed from the surface before the soil sample was extracted. Once in the laboratory, the samples were dry-sieved through a 1 mm mesh and labeled appropriately.

Soil analyses were carried out at the Laboratory of Instrumental Techniques of the University of León. Chemical (pH, organic matter, total and extractable nitrogen, C/N ratio, soil exchangeable macronutrients P, $K^+$, $Ca^{2+}$, $Mg^{2+}$), physical (moisture content) and biochemical properties (urease, phosphatase and β-glucosidase activity) were determined. These variables were selected as they are commonly used in studies related to the effects of vegetation treatments on soil properties [23,33,34].

For the conditioning of the samples and the determination of nitrogen and organic matter, a portion of soil was ground with a ball mill to yield small particles (0.2 mm). The pH was measured in a soil/water suspension (ratio 1:2.5) ratio with a pH meter. Organic matter (%) was determined by the Walkley–Black method [35]. The total N (%) was determined by the Kjeldahl method [36]. The available phosphorus was determined by the Olsen method [37], with extraction by 0.5 M $NaHCO_3$ pH 8.5 and reading by molecular spectrometry in a UV/VIS spectrophotometer. The cation (calcium, magnesium and potassium) concentrations were measured by ICP-OES, after extraction in 1N AcONH4 pH 7.

The soil moisture content was determined by the gravimetric method [38], i.e., a sample of soil was weighed before and after drying in an oven at 105 °C and the moisture content was calculated on a dry basis.

The β-glucosidase (EC 3.2.1.21; β-D-glucoside glucohydrolase) and acid phosphatase (phosphatase: EC 3.1.3.2; phosphate monoester phosphohydrolase) activity was determined following the procedure described by [39]. The urease (urease: EC 3.5.1.5; urea amidohydrolase) activity was determined according to [40].

*2.5. Statistical Analysis*

The statistical analysis was performed using the R open-source software v4.3.2 [41]. Two types of analysis were conducted. First, the effects of the two treatments (burning and shredding) on the soil properties at the different sampling times (i.e., post_0, 1 year and 2 years) were compared. In addition, tests were carried out to detect any significant differences in the soil properties before the treatments (pre_0) and two years after the treatments, in order to determine whether any substantial changes in these properties had occurred relative to the pre-management scenario.

For each soil type (related to each vegetation community), the effects of the treatment and time since treatment on the soil parameters were analyzed by a linear mixed effects model. Both the treatment and the time since treatment were included as fixed factors. The interaction between both factors was also taken into account. The plot was considered to have a random effect on the intercept in the model. A significance level ($\alpha$) of 5% was applied. The "lmer" function of the "lme4" package of R [41] was used to fit the mixed model using restricted maximum likelihood (REML), and the Kenward–Roger approximate F-test was used to estimate fixed effects [42].

The results are reported separately for each soil type, i.e., for each type of plant community studied (shrubland and heathland).

## 3. Results

The changes in the soil properties in the gorse shrubland and heathland plots are shown in Figures 2 and 3, respectively, for both treatments and the four sampling times: 0-pre, 0-post, 1 year and 2 years.

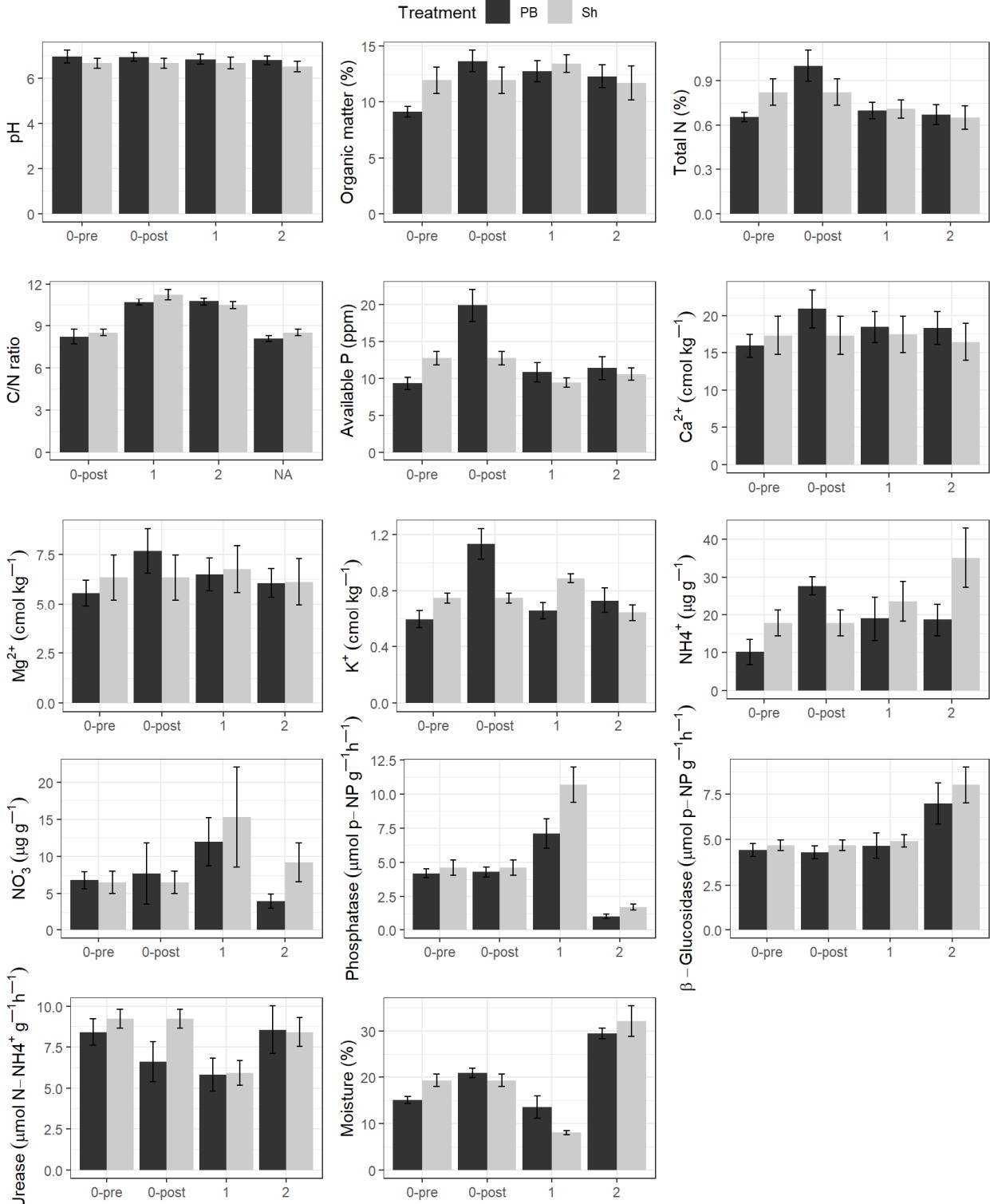

**Figure 2.** Changes in soil properties in gorse shrubland plots for the prescribed burning (PB) and shrub shredding (Sh) treatments: immediately before and after treatment (0-pre and 0-post, respectively) and one year (1) and two years (2) after treatment.

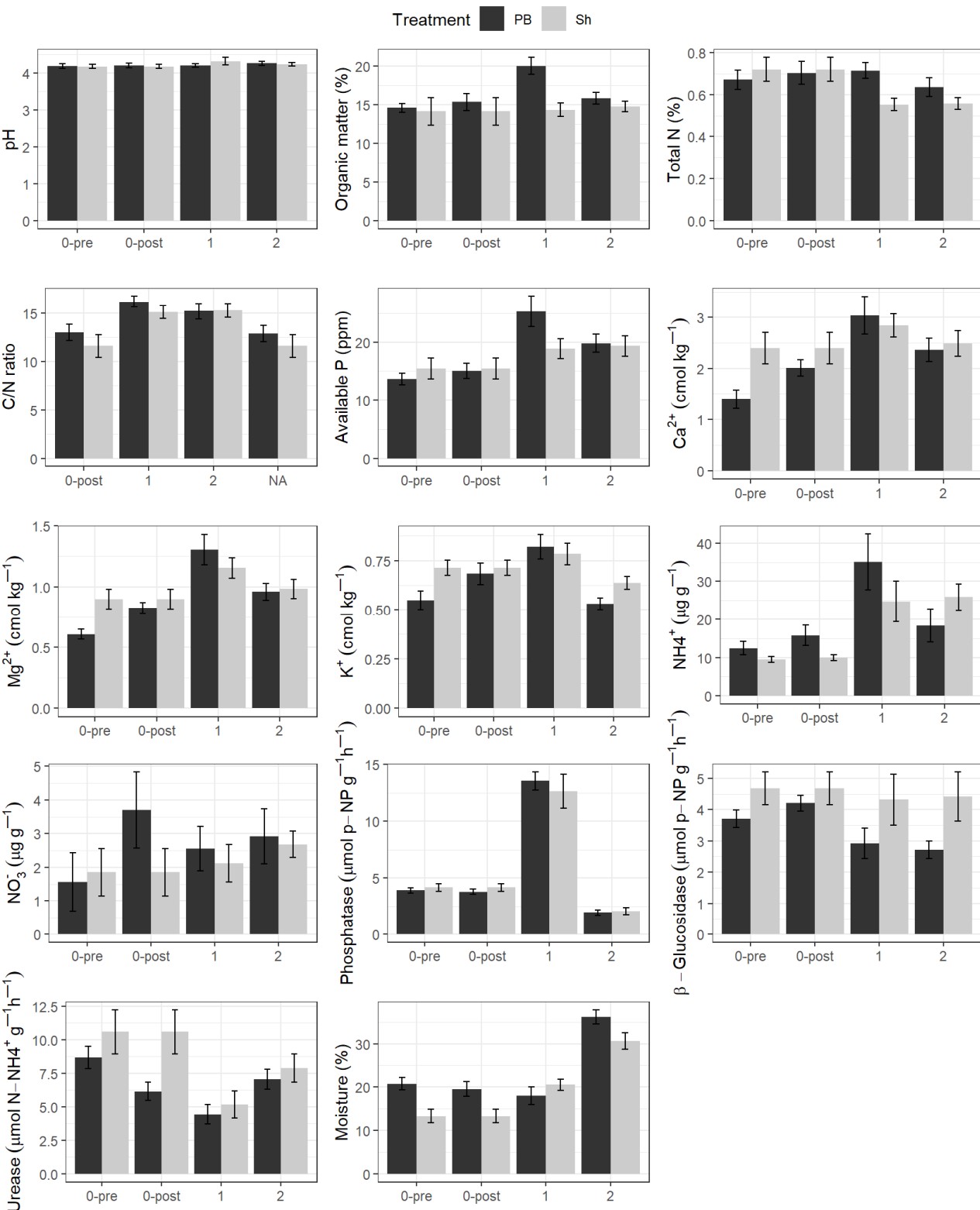

**Figure 3.** Changes in soil properties in heathland plots for the prescribed burning (PB) and shrub shredding (Sh) treatments: immediately before and after treatment (0-pre and 0-post, respectively) and one year (1) and two years (2) after treatment.

### 3.1. Effects of Vegetation Treatments and Sampling Dates on Soil Properties

The linear mixed model revealed significant effects of the treatment, sampling date and, to a lesser extent, the interaction between the time and treatment factors in most of the soil properties analyzed (Table 3). Overall decreases in total N and phosphorus were observed over time, whereas the opposite trend was observed for the C/N ratio. For potassium, $NH4^+$ and soil moisture, the sampling time was also significant, but no regular pattern was observed. Regarding microbial activity, while the values of urease and β-glucosidase peaked after two years, the opposite trend was observed for phosphatase. Six soil properties were affected by the treatments. Total N and available P were consistently higher after prescribed burning, while the opposite was observed for phosphatase activity, which was consistently higher after shredding. The effects of the treatments on the potassium content, $NH4^+$ and soil moisture were strongly influenced by the sampling time (Table 3 and Figure 2).

**Table 3.** Results of the mixed model analysis comparing the effects of the prescribed burning and shrub shredding treatments and sampling dates (post_0, 1 and 2 years) on the soil properties in gorse shrublands. Significant effects are highlighted in bold. *** indicates $p < 0.001$, ** $p < 0.01$, * $p < 0.05$.

| Soil Property | Time | | Treatment | | Treatment × Time | |
|---|---|---|---|---|---|---|
| | F-Value | *p*-Value | F-Value | *p*-Value | F-Value | *p*-Value |
| pH | 1.3305 | 0.8562 | 5.5411 | 0.1362 | 0.203 | 0.9035 |
| Organic matter | 5.6079 | 0.2304 | 3.9462 | 0.2673 | 3.0079 | 0.2223 |
| Total N | 39.703 | **$4.987 \times 10^{-8}$ ***** | 10.586 | **0.01419 *** | 6.921 | **0.03141 *** |
| C/N ratio | 59.974 | **$2.937 \times 10^{-12}$ ***** | 3.241 | 0.3559 | 2.500 | 0.2865 |
| Available P | 31.183 | **$2.809 \times 10^{-6}$ ***** | 12.415 | **0.006089 **** | 6.0566 | 0.0584 |
| Ca | 3.7625 | 0.4391 | 6.364 | 0.09518 | 1.3868 | 0.4999 |
| Mg | 6.047 | 0.1957 | 3.5746 | 0.3112 | 2.946 | 0.2292 |
| K | 29.791 | **$5.397 \times 10^{-6}$ ***** | 21.131 | **$9.887 \times 10^{-5}$ ***** | 19.517 | **$5.78 \times 10^{-5}$ ***** |
| $NH4^+$ | 11.585 | 0.05072 | 11.054 | **0.01144 *** | 10.181 | **0.006153 **** |
| $NO_3^-$ | 7.0128 | 0.1352 | 4.2107 | 0.2396 | 3.7576 | 0.1528 |
| Phosphatase | 94.522 | **<$2.2 \times 10^{-16}$ ***** | 14.945 | **0.001864 **** | 4.4457 | 0.1083 |
| β-Glucosidase | 22.868 | **0.0001345 ***** | 2.559 | 0.521 | 0.0932 | 0.9545 |
| Urease | 12.239 | **0.01566 *** | 5.6764 | 0.1285 | 2.9812 | 0.2252 |
| Moisture | 81.608 | **<$2.2 \times 10^{-16}$ ***** | 8.9216 | **0.03035 *** | 7.5816 | **0.02258 *** |

For the heathland plots, the sampling date and, to a lesser extent, the treatment factor had significant effects on most of the soil properties analyzed (Table 4). A general decrease over time was only observed for total N, whereas the opposite trend was observed for the C/N ratio and P. The sampling time also significantly affected the cations analyzed (calcium, magnesium and potassium), $NH4^+$ and acid phosphatase, with the maximum values reached one year after treatment. The opposite trend was observed for urease, with the minimum values occurring at this time. Only four soil properties were affected by the treatments: organic matter, total N and available P were always higher after prescribed burning, whereas the opposite was observed for β-glucosidase and urease activity, which were always higher after shrub shredding. The effects of the treatments on the soil moisture content and ammonium ($NH4^+$) were largely influenced by the sampling time (Table 4 and Figure 3).

**Table 4.** Results of the mixed model analysis comparing the effects of the prescribed burning and shrub shredding treatments and sampling dates (post_0, 1 and 2 years) on the soil properties in the heathland plots. Significant effects are highlighted in bold. *** indicates $p < 0.001$, ** $p < 0.01$, * $p < 0.05$.

| Soil Property | Time | | Treatment | | Treatment × Time | |
|---|---|---|---|---|---|---|
| | F-Value | *p*-Value | F-Value | *p*-Value | F-Value | *p*-Value |
| **pH** | 4.2317 | 0.3756 | 2.1127 | 0.5493 | 1.9987 | 0.3681 |
| **Organic matter** | 10.943 | 0.05721 | 13.768 | **0.003239 *** | 5.869 | 0.05316 |
| **Total N** | 14.938 | **0.004831 *** | 10.013 | **0.01846 *** | 5.44333 | 0.0661 |
| **C/N ratio** | 24.605 | **$6.038 \times 10^{-5}$ *** | 3.4781 | 0.3236 | 1.3686 | 0.5044 |
| **Available P** | 30.221 | **$4.412 \times 10^{-6}$ *** | 6.8117 | 0.07815 | 4.3548 | 0.1133 |
| **Ca$^{2+}$** | 14.771 | **0.005201 *** | 2.6334 | 0.4517 | 2.2021 | 0.3325 |
| **Mg$^{2+}$** | 21.474 | **0.000255 *** | 2.3773 | 0.4979 | 2.2971 | 0.3171 |
| **K$^+$** | 21.605 | **0.0002402 *** | 3.3061 | 0.3468 | 2.5205 | 0.2836 |
| **NH4$^+$** | 20.374 | **0.0004213 *** | 8.1417 | 0.05317 | 7.9808 | **0.01849 *** |
| **NO$_3{}^-$** | 1.9453 | 0.7458 | 1.6253 | 0.6537 | 1.5221 | 0.4672 |
| **Phosphatase** | 125.83 | **$2.2 \times 10^{-16}$ *** | 1.5105 | 0.6798 | 1.4945 | 0.4737 |
| **β-Glucosidase** | 8.9473 | 0.06243 | 14.945 | **0.001864 *** | 4.4457 | 0.1083 |
| **Urease** | 21.908 | **0.0002091 *** | 8.5087 | **0.03659 *** | 2.267 | 0.2315 |
| **Moisture** | 71.207 | **$1.262 \times 10^{-14}$ *** | 14.307 | **0.002516 *** | 9.4396 | **0.008917 *** |

### 3.2. Medium-Term Effects of Vegetation Treatments on Soil Properties

In the gorse shrubland plots, for the two-year period (2016–2018), significant increases in organic matter, the C/N ratio, NH4$^+$, β-glucosidase and soil moisture were observed. The values of the latter four variables increased for both the prescribed burning and shrub shredding treatments, whereas the organic matter only increased in the plots treated by prescribed burning. Moreover, there were no significant effects on the C/N ratio, β-glucosidase or soil moisture.

On the other hand, the values of total N, NO$_3{}^-$ and phosphatase activity decreased significantly during the study period. For these soil properties, except phosphatase activity, the effect of the treatment was largely mediated by the treatment x time interaction (Figure 2 and Table 5).

**Table 5.** Results of the mixed model analysis comparing the effects of the prescribed burning and shrub shredding treatments and sampling dates (post_0, 1 and 2 years) on the soil properties in the gorse shrubland plots. Significant effects are highlighted in bold. *** indicates $p < 0.001$, ** $p < 0.01$, * $p < 0.05$.

| Soil Variable | Time | | Treatment | | Treatment × Time | |
|---|---|---|---|---|---|---|
| | F-Value | *p*-Value | F-Value | *p*-Value | F-Value | *p*-Value |
| **pH** | 1.111 | 0.5738 | 3.9604 | 0.138 | 0.0041 | 0.9491 |
| **Organic matter** | 5.8721 | **0.01538 *** | 8.0437 | **0.01792 *** | 5.8721 | **0.01538 *** |
| **Total N** | 9.0524 | **0.01082 *** | 8.6048 | **0.01354 *** | 5.6319 | **0.01764 *** |
| **C/N ratio** | 49.138 | **$2.13 \times 10^{-11}$ *** | 2.4952 | 0.2872 | 2.3875 | 0.1223 |
| **Available P** | 5.901 | 0.05231 | 9.4861 | **0.008712 *** | 5.8316 | **0.01574 *** |
| **Ca$^{2+}$** | 2.601 | 0.2724 | 2.1913 | 0.3343 | 2.1543 | 0.1422 |
| **Mg$^{2+}$** | 0.5836 | 0.7469 | 1.1734 | 0.5562 | 0.5033 | 0.478 |
| **K$^+$** | 4.1099 | 0.1281 | 4.3304 | 0.1147 | 4.0483 | 0.05422 |
| **NH4$^+$** | 6.5419 | **0.03797 *** | 9.9933 | **0.00676 *** | $5 \times 10^{-4}$ | 0.9816 |
| **NO$_3{}^-$** | 6.1477 | **0.04624 *** | 7.6636 | **0.02167 *** | 5.9515 | **0.0147 *** |
| **Phosphatase** | 56.613 | **$5.089 \times 10^{-13}$ *** | 10.899 | **0.004298 *** | 4.3868 | **0.03622 *** |
| **β-Glucosidase** | 17.65 | **0.000147 *** | 1.4946 | 0.4736 | 0.1988 | 0.6557 |
| **Urease** | 0.8235 | 0.6625 | 0.7103 | 0.7011 | 0.1073 | 0.7432 |
| **Moisture** | 41.5 | **$9.735 \times 10^{-10}$ *** | 5.2371 | 0.07291 | 0.2763 | 0.5991 |

In the heathland plots, significant increases in the C/N ratio, available phosphorus, $Ca^{2+}$, $Mg^{2+}$, ammonium ($NH4^+$) and soil moisture were observed after two years. The values of the first two variables increased after both types of treatment. However, only $NH4^+$ increased in the plots treated by shredding and $Ca^{2+}$ and $Mg^{2+}$ in the plots treated by burning. For soil moisture, a significant treatment effect was also observed, with the highest values in the plots treated by prescribed burning.

The total nitrogen and phosphatase values increased after two years for both treatments. By contrast, significant differences in potassium and β-glucosidase were seen depending on the treatment applied, with higher values in the plots treated by shredding than in those treated by prescribed burning (Figure 3 and Table 6).

**Table 6.** Results of the mixed model analysis comparing the effects of the prescribed burning and shredding treatments and sampling date (0-post, 1 and 2 years) on the soil properties in the heathland plots. Significant effects are highlighted in bold. *** indicates $p < 0.001$, ** $p < 0.01$, * $p < 0.05$.

| Soil Variable | Time | | Treatment | | Treatment × Time | |
|---|---|---|---|---|---|---|
| | F-Value | *p*-Value | F-Value | *p*-Value | F-Value | *p*-Value |
| pH | 3.1956 | 0.2023 | 0.396 | 0.8204 | 0.0549 | 0.8147 |
| Organic matter | 0.6301 | 0.7297 | 0.908 | 0.6348 | 0.0887 | 0.7658 |
| Total N | 9.3447 | **0.00935 **** | 2.6658 | 0.2637 | 2.6061 | 0.1065 |
| C/N ratio | 14.31 | **0.0007811 ***** | 1.5622 | 0.4579 | 0.8843 | 0.347 |
| Available P | 22.461 | **$1.327 \times 10^{-5}$ ***** | 1.7457 | 0.4178 | 1.3788 | 0.2403 |
| Ca | 13.265 | **0.001317 **** | 14.298 | **0.0007857 ***** | 6.0884 | **0.01361 *** |
| Mg | 14.32 | **0.007769 **** | 10.04 | **0.006606 **** | 4.5587 | **0.0327 *** |
| K | 2.7467 | 0.2533 | 14.325 | **0.0007749 ***** | 0.8271 | 0.3631 |
| $NH_4^+$ | 13.553 | **0.00114 **** | 4.9405 | **0.0845 *** | 4.0914 | **0.0431 *** |
| $NO_3^-$ | 5.7085 | 0.0576 | 2.5223 | 0.2833 | 0.6753 | 0.4112 |
| Phosphatase | 37.65 | **$6.67 \times 10^{-9}$ ***** | 0.2979 | 0.8616 | 0.006 | 0.9384 |
| β-Glucosidase | 4.7533 | 0.09286 | 7.4985 | **0.02354 *** | 0.76 | 0.3833 |
| Urease | 4.4464 | 0.1083 | 1.216 | 0.5242 | 0.0486 | 0.8255 |
| Moisture | 51.52 | **$6.496 \times 10^{-12}$ ***** | 15.022 | **0.0005471 ***** | 0.4111 | 0.5214 |

## 4. Discussion

This study's findings provide new insights into how vegetation management treatments affect soil characteristics. The main contributions of the study are derived from the comparison of the medium-term effects of two alternative treatments in soils of a different pH. Numerous studies have investigated how soil properties are affected by prescribed burning [19,21,33,43–48] and mechanical treatments [49–51], but studies comparing different treatments are scarce [23]. Likewise, to our knowledge, no other studies have simultaneously analyzed such effects in soils of a different pH. We also studied the medium-term effects, which are usually more informative than short-term effects in terms of recommending the best techniques for the management of vegetation.

The exact management history of the study sites (type, frequency, intensity) is not known, and this lack of information could affect the interpretation of the results. Nevertheless, due to the long history of traditional livestock management in the Natural Park of Babia y Luna, it can be assumed that all sites are managed by transhumant sheep grazing with moderate stocking rates and short seasonal use. Woody plants are probably burned or shredded when they encroach upon grazing areas, but the exact dates of such treatments are not known for each plot.

As a result of the analysis of a large number of soil variables, the discussion of the results is not straightforward, and we focus on explaining the main patterns observed in the main soil properties.

The pH values did not undergo significant changes either after the treatments or when comparing the situation before and two years after the treatments (Tables 3–6 and Figures 2 and 3). In the prescribed burning treatment, higher pH values were expected two years after the treatments due to the deposition of alkaline ash [52,53]. The absence

of significant changes is consistent with the results reported in several studies in different vegetation communities worldwide [26,54–57], which indicate negligible variations in pH in response to low-intensity burning. Ref. [34] also did not observe any significant difference between prescribed burned and control plots 4 months after a fire in Cantabrian heathlands, close to the present study area. The lack of a significant increase in the soil pH could be explained by the incomplete incorporation of the ash in the soil and the relatively low temperatures reached during the prescribed burning [58–60].

Although there were no differences in the organic matter content after the treatments, an increase was observed after two years relative to the original situation in the prescribed burning treatment; the increase was statistically significant in the gorse shrubland soils. These results are consistent with those reported by [33,61] in their comprehensive literature reviews. In these studies, conducted in both woodland and shrubland ecosystems, the researchers observed that fire had very variable effects on the soil organic matter content that depended on several factors, including the fire severity. Low-severity fires generally have little effect and can even led to an increase in organic matter, as observed by [62] in Mediterranean shrublands in Portugal, with a similar vegetation composition to the shrublands under study here. The increase in organic matter in this case may be related to rapid colonization by plant species after burning [63,64], as the roots of herbaceous plants contribute to the accumulation of organic matter in the soil [64,65]. In prescribed burning, the combustion of soil organic matter occurs from approximately 200 °C [66], i.e., at much higher temperatures than reached in the soil in this study.

The total N decreased significantly in the soils under both plant communities in the medium term and relative to both the pre-treatment (pre_0) and post-treatment (post_0) situations. These results are also consistent with previous findings in Mediterranean shrublands in Spain located on different types of soil [21,67], for which an increase in N immediately after the prescribed burning and the return of the total N values to pre-treatment levels or even lower levels in the medium term were reported. The decreases in total N associated with prescribed burning may have been caused by N volatilization and the loss of fly ash [68,69].

The medium-term decrease in organic matter may have been due to the exposure to heat and solar radiation caused by the treatments, which can accelerate the microbial decomposition of soil organic matter. This may result in the temporary release of nitrogen in the form of ammonium ($NH4^+$), which may subsequently be leached or transformed into gaseous forms that are released into the atmosphere in the form of nitrogen and ammonium oxides. As a result, after both treatments, the ammonium concentrations were higher than the pre-treatment values (Figures 2 and 3, Tables 5 and 6) [20,70]. Nitrate ($NO_3^-$) may also increase after burning due to mineralization processes, and it can be absorbed by plants or microorganisms or lost by denitrification and/or leaching [20,71]. The increase may also have occurred due to decreased uptake by plants [72].

According to [73], for burned plots located in vegetation dominated by herbaceous plants and *Quercus* trees, the nitrogen content in burned soils tends to increase or decrease over time according to the abundance of perennial herbaceous plants. In burns carried out in different Mediterranean plant communities, refs. [63,74] observed a general decrease in nitrogen concentrations after six months, relative to the concentrations measured immediately after the fire, which was attributed to N consumption during the germination of some species that appear in post-fire conditions. By contrast, ref. [34] did not observe any significant differences in N after prescribed burning in heathlands close to the study area.

The C/N ratio increased significantly two years after the treatments, relative to both the pre-treatment and post-treatment situations. This increase was the consequence of both the reduction in total N and the maintenance of or increase in organic matter content (related to soil organic carbon) after the treatments. In low-intensity and -severity fires, semi-pyrolyzed plant matter can accumulate on the surface, thus increasing the total carbon content of the soil [75,76], as shown by several authors for burns carried out in pine forests. An increase in carbon concentrations may also occur immediately or shortly after prescribed

burns due to the incomplete burning of organic matter caused by low temperatures in the upper soil layer or at the top of the mineral soil [77], or due to the incorporation of unburned or partially burned material into the soil in different vegetation communities worldwide [44,60,78–81].

The available P decreased over time after both vegetation treatments in the shrubland plots, whereas it increased significantly in the heathland plots. In a study of communities dominated by *Erica* sp. and *Ulex* sp., ref. [61] also found that the patterns of change for this soil property after a fire varied significantly according to the initial soil reaction. Our findings show that the available P in heathlands increased substantially after both treatments and remained higher than the pre-treatment situation two years after the experiment. This increase, in the case of prescribed burning, may be due to the increase in temperature, capable of mineralizing organic phosphorus, as mentioned by some authors for burns carried out in very different vegetation communities [82,83], and also to the release of phosphorus previously fixed in the burned biomass, which is subsequently deposited together with ash, as reported for burns carried out in heathlands [27]. These results are consistent with those obtained in acidic heathland soils in NW Spain [23,84], although the magnitude and duration of the effects were different. According to [23], in plots treated by prescribed burning, the available P values were higher at 6 months and two years (104% and 43%, respectively) than in untreated plots, but no significant differences were observed after four years. According to [34], in heathlands in the Cantabrian Mountains, the available P concentrations were higher (although the difference was not statistically significant) in burned plots than in control plots in samples taken in the first year after treatment; however, in the second year, the P values were very similar to those found in untreated areas. It should be noted that P is one of the most limiting nutrients in *Calluna vulgaris* heathland [85].

The increase in available P in the plots treated by shredding may be attributed to the release of nutrients, including phosphorus, to the soil as the plants decompose; [34] also reported that the P content was 39% higher in shredded plots than in untreated plots 6 months after treatment.

The concentrations of the cations $Ca^{2+}$, $Mg^{2+}$ and $K^+$ in the heathland plots increased notably one year after the treatments; however, the change seemed to be transient as the concentrations subsequently decreased, returning to the initial or even lower levels, as observed in previous studies in different vegetation communities [33,44]. The decrease in the concentrations of these cations may have been the result of leaching processes and/or plant uptake after regrowth following burning or shredding, as shown by different authors after prescribed burns in the understories of Mediterranean forests and in open shrublands [44,86].

Regarding enzymatic activity, the values were generally higher in the plots treated by shredding than those treated by burning. This pattern was common for both vegetation communities (Figures 2 and 3 and Tables 3–6).

In the gorse shrubland plots, two years after the treatments, the acid phosphatase had decreased, β-glucosidase had increased, and urease had not undergone any significant changes relative to the pre-treatment situation; the treatment did not significantly affect any of the three variables. For heathland plots, only acid phosphatase followed the above-mentioned pattern, whereas β-glucosidase did not vary over time but did vary with treatment, and urease decreased over time. The variation in results depending on the soil pH and treatment emphasizes the fact that soil enzymatic activity can be affected differently according to the site and treatment [23]. For example, decreases in acid phosphatase and β-glucosidase activity have been observed in shrublands in NW Spain [87]. However, other authors have reported the absence of changes in acid phosphatase activity following a low-severity fire in gorse [88] and increases in acid phosphatase and β -glucosidase activity after prescribed fires in Mediterranean shrublands [89].

The decrease in acid phosphatase over time is consistent with previous findings in acidic soils [90] from different forest ecosystems, showing that urease and β-glucosidase

are more resistant than phosphatase, which decreased significantly after burning, even after low-intensity burning. Similarly, in a study in heathlands in NW Spain [23], the phosphatase activity was lower in plots treated by prescribed burning than in control plots over a 4-year-period and for most sampling dates after treatment application.

Regarding β-glucosidase activity, in ref. [23], in a study carried out on acidic soils, similar results to those obtained for heathlands in the present study were observed: β-glucosidase was lower in prescribed burned plots than in control plots for the overall study period, although the decrease was more moderate than for phosphatase.

On the other hand, burning can affect the soil microbial community, although, in low-severity fires, the high microbial richness can affect the response of ecosystem multifunctionality [84]. Although this effect is more marked in shrublands, it also occurs in heathlands.

The soil moisture remained relatively consistent from 0-pre to year 1 across both vegetation communities and treatments. Nevertheless, a sharp increase was then observed in year 2 in all sites. This increase reflects weather patterns, as the sampling in 2018 was carried out shortly after an episode of strong storms in the area. Regardless of the sampling date, the soil moisture content tended to be higher in the plots treated by prescribed burning than in those treated by shredding. These results are consistent with previous findings in *Calluna vulgaris* heathlands in the Cantabrian Mountains, i.e., consistently higher moisture content in burned plots than in control plots [34]. Other studies have also detected higher moisture content in samples after prescribed burning treatments, carried out in different plant communities worldwide, relative to control samples [91], while others did not find any significant differences [81,92]. In the present study, this finding may be explained by the low intensity of the fire, as the surface layer of organic matter did not burn completely, which facilitated water retention in the soil, as reflected in some studies on heathlands in the Cantabrian mountains [34]. The absence of woody vegetation also reduces water loss through transpiration in the deeper soil layers, as shown by some authors after prescribed burns in shrublands and forests [81].

Fire has been shown to affect the physical, chemical and biological properties of soils, depending on the fire severity and frequency [34,61,92]. The impact of individual fires on soil properties has been widely studied, and prescribed burning has generally been found to have a limited effect on soil, as long as it is carried out correctly, avoiding high soil temperatures being reached. The results of the present study confirm these previous findings, in both acidic heathland soils and alkaline gorse shrubland soils. Nevertheless, as fuels accumulate over time, fuel reduction techniques such as prescribed burning must be applied periodically to maintain the fuel hazard at tolerable levels, which may exacerbate the impact of the fire on the soil. In the case of repeated prescribed fires, the studies reviewed by [48] show that physical properties are generally less affected than chemical properties, which are, in turn, less affected than microbiological properties. Nonetheless, ref. [48] also found that the effects were highly variable and rather inconsistent, with a number of soil properties showing contrasting responses.

Additionally, vegetation and the soil response to fire are inextricably linked [93]. The observed changes in chemical properties are expected to affect the vegetation composition and grassland structure [94–96]. Although there is little evidence that the use of prescribed burning to improve the grazing quality in mountain pastures has had much effect on the plant species richness and composition in the long term [97], conducting additional research focused on analyzing plant diversity and herbaceous vegetation could be beneficial in the present case. Such studies could provide valuable information regarding how nutrients may influence the vegetation composition, encouraging more palatable species for livestock. At the same time, such research efforts could contribute to the development of management plans aimed at reducing wildfire risks [63].

### 5. Conclusions

The present findings provide further information on the medium-term effects of two vegetation treatments, shredding and prescribed burning, on soils under different shrub vegetation communities. Both treatments have been widely used to manage woody plant encroachment in grasslands and to maintain open ecosystems for grazing in the study area.

In the soils of both shrub communities (gorse shrubland dominated by *Genista hispanica* subsp. *occidentalis* and heathland dominated by *Calluna vulgaris*), a decrease in total N and acid phosphatase and an increase in the C/N ratio were observed after 2 years, independently of the vegetation treatment applied. The available P concentrations varied according to the soil pH, and they only increased in the acidic soils in the heathland plots. Regarding the effects of the vegetation treatments, the total N and available P were higher after the prescribed burning treatment, and the enzymatic activity was generally higher after the shredding treatment.

Overall, our findings suggest that vegetation management strategies, i.e., prescribed burning and shredding, do not have drastic effects on either chemical or soil enzymatic properties. Therefore, from this point of view, neither treatment can be recommended over the other. From a practical perspective, other considerations apart from the effects of management treatments on soil properties should be also considered when selecting the most appropriate strategy (e.g., soil erosion, effects on fauna, etc.). However, the application of prescribed burning is already hampered by difficulties in determining the windows of suitable meteorological conditions within which the operational objectives can be achieved without damaging the ecosystem, and these windows are shifting due to climate change.

**Author Contributions:** R.M.C.: conceptualization, methodology, formal analysis, investigation, writing—original draft preparation; F.C.-D.: conceptualization, methodology, formal analysis, investigation, writing—review and editing; L.V.: conceptualization, methodology. All authors have read and agreed to the published version of the manuscript.

**Funding:** This research received no external funding.

**Informed Consent Statement:** Not applicable.

**Data Availability Statement:** The data that support the findings of this study are available from the corresponding author upon reasonable request.

**Acknowledgments:** This study was carried out thanks to the collaboration of the Territorial Environmental Service of León (Junta de Castilla y León), which financed the soil analyses and provided the fire-fighting equipment for the burns (Romeo 8.3 and Charlie 8.3) and the clearing (Romeo 8.3). We also thank the Forest Fire Laboratory of the University of Cordoba (LABIF-UCO) and the MasterFuego students Claudia Muñoz, Luis López, Victor Riera and Federico Gallardo for their helpful collaboration in conducting the prescribed burns.

**Conflicts of Interest:** The authors declare no conflicts of interest.

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
