# Peer review of "Medium-Term Comparative Effects of Prescribed Burning and Mechanical Shredding on Soil Characteristics in Heathland and Shrubland Habitats: Insights from a Protected Natural Area"

_fire, doi:10.3390/fire7050160_

Round 1

Reviewer 1 Report

Comments and Suggestions for Authors

This study evaluates the effect of two different land management treatments on soil characteristics in two vegetation communities in the NW of Spain. The manuscript is well written, and the study is quite relevant; gaining a better understanding of the immediate and medium-term consequences of different land management practices is critical to reduce fire risk without compromising key soil characteristics. I have a few comments I would like the authors to address before I can recommend this manuscript for publication.

Although many of the soil properties investigated were significantly affected by treatment type and time since treatment, the authors concluded that prescribed burning and shredding do not have substantial effects on soil characteristics. This can be inferred from the figures, but it would be good to report marginal and conditional R2 of the mixed-effect models to support this statement.

Prescribed burn and shrub shredding already take place at the Babia y Luna Natural Park (LINE 55). What is the management history of the six study sites? Were they burnt and/or shredded before the study took place? If so, how often and when was the last time they were treated? This information could affect the interpretation of the results.

The authors acknowledge that the large amount of data included in this study resulted in a non-straightforward discussion of the results (LINE 314), which is understandable. However, the current structure makes it difficult to understand the ecological implications of the authors’ findings and the relevance of this work beyond the management of that specific natural park. For instance, citations from i) local studies, ii) international studies of comparable vegetation communities, and iii) studies on different vegetation communities are presented without distinguishing them. It would help to better distinguish between these comparisons. If the authors find it appropriate, it could even be used as an organising principle to restructure the Discussion.

Soil moisture is relatively consistent from 0-pre to year 1 across both veg communities and treatments (with the exception of shrub shredding in year 1). A sharp increase is then observed in year 2 within all sites. Is it possible that this simply reflects weather patterns? Was rainfall particularly abundant in year 2 prior to soil sampling? Regardless, this point should be addressed in the Discussion, to either exclude the effect of rainfall or account for it.

Specific remarks

LINE 25. “…important effects on chemical and soil enzymatic properties”. Important short- and medium-term effects.

LINE 174. Briefly describe why those physical, chemical, and biochemical properties were chosen.

Figure 1. Since the study sites are all located in the NW portion of the Babia y Luna Natural Park, I wonder if it would be more informative to zoom in on the study area.

Figures 2-5. Why was ‘0-pre’ not included in figures 2 and 3? It would remove two figures (Figs. 4 and 5) and make it easier for the reader to compare temporal trends. Also, boxplots might be more informative than bar charts for this kind of data.

Tables 3-6. Significant effects are all indicated by a single asterisk, regardless of the significance level. Why was that choice made? Typically, significance levels are flagged using the following code: *** indicates P < 0.001, ** P < 0.01, *P < 0.05.

LINE 225. “Only six soil properties…” That is nearly half of the properties evaluated.

LINE 442. The potential long-term effects of repeated treatments on soil and plant communities should also be explicitly discussed. Some of the literature already cited by the authors addresses this issue (e.g., 47, 72, 73).

LINE 464. This statement cannot be made based on the results of this study. As a single treatment was investigated, compounding effects could not be addressed (see comment above).

LINE 468. It is worth mentioning that prescribed burn windows are shrinking due to climate change.

Reviewer 2 Report

Comments and Suggestions for Authors

This paper addresses an information gap about effects of prescribed fire vs. mastication (or ‘shredding’) on two soil types in Spain. There is much painstakingly collected and valuable soils data here. I had questions about the choices in data analysis and presentation. It seemed odd to have one set of figures and tables presenting immediately post, 1 year post, and 2 year post treatment data, and another set of figures and tables presenting pre-treatment and 2-year post treatment data. It would be clearer and more efficient to have just one set of figures and tables comparing pre, immediate post, 1 year post, and 2-year post treatment. Specific contrasts, such as comparing pre- vs. 2 year data, could be done with the ‘contrast’ statement in lmer. But lmer should automatically present that information. I was also puzzled to find tables with Chi-square values – I would have expected to see F values instead. It would be helpful to have a line in the text explaining this choice. In general, though, it is useful to have this information about the relative lack of impact of different treatments on the soil in this system.

Other comments

Table 1: Explain how these data were collected. Is there a reason why statistical significance of difference between communities is not provided? Delete last line of table – Fuel Model is not a mean value.

L349. Increases in NO3 after burning may occur because of decreased uptake by plants. See Ficken and Wright (2017).

L372. It is incorrect to refer to the pre-treatment data as a control. There is no control in this experiment – just two treatments.

EDITORIAL COMMENTS

L 97. Use lat/long or state the UTM zone

L 102. I don’t know what the Natura 2000 Network is, nor ZEC or ZEPA. If these are important, spell them out and explain them. If not important, delete.

L114. Provide the taxonomic authority when a species is introduced.

L111-121. It is easier for the reader if you maintain ideas in the same order. Thus if you introduce alkaline before acidic soils, it is helpful if you describe the plant species on alkaline soils before the species of acidic soils.

L 116 Check spelling of Juniperus

L119. See comments on line 102. What is Habitats Directive 92/43/EEC?

L124. Describe treatments before describing plot layout.

L 125 Delete ‘that’

L 128-129. I believe that ‘two’ should be deleted – it makes it sound as though two community types on each soil type were sampled, but the subsequent description implies that only one community type (the most characteristic one)was sampled on each soil type.

L137. Delete the first sentence of this paragraph – it is awkward and wasteful of space to use an entire sentence to draw attention to a table. Instead, make an assertion about the data then back up the assertion putting the table (or fig) number in parentheses. Here, ‘(Table 1)’ should be placed after the words ‘bare ground’ at the end of what is currently the second sentence.

L143. (Table 1) Left-justify the contents of the first column. That helps to distinguish them from the contents of the second and third columns.

L 156. Use the numeric form of ‘one’

L212-213. See comment on L137.

Fig. 2, 3, and table 3. C/N ‘rate’ should be changed to C/N ‘ratio’.

L309. It seems the assertion is being made that there is little information about mechanical treatment effects on soil properties, which is incorrect.

L310. Check spelling of ‘simultaneously’

L367. Earlier, this measurement is referred to as ‘available P’. Now it is referred to as ‘assimilable P’. It would be better to maintain consistent terminology.

REFERENCES:

Check references for consistent capitalization. In some references (e.g., [14,18]), the first letter of every word is capitalized but in most it is not.

Round 2

Reviewer 1 Report

Comments and Suggestions for Authors

The authors have addressed all points I made. I can now recommend this manuscript for publication.